# Effects of Heavy Isotopes (^2^H_1_ and ^18^O_16_) Depleted Water Con-Sumption on Physical Recovery and Metabolic and Immunological Parameters of Healthy Volunteers under Regular Fitness Load

**DOI:** 10.3390/sports9080110

**Published:** 2021-08-11

**Authors:** Zaira Kharaeva, Tamara Hokonova, Jannet Elmurzaeva, Irlana Dzamihova, Wolfgang Mayer, Chiara De Luca, Ilya Trakhtman, Liudmila Korkina

**Affiliations:** 1Department of Microbiology, Virology, and Immunology, Kabardino-Balkar Berbekov’s State University, 176 Chernishevskogo St., 360000 Nal’chik, Russia; irafe@yandex.ru (Z.K.); tamy07@inbox.ru (T.H.); jannet.elmurzaeva@yandex.ru (J.E.); 2Fitness Centre “S-Club”, 36 Kuliev Pr., 360030 Nal’chik, Russia; irlanchik@mail.ru; 3R&D Department, MEDENA AG, 16 Industriestrasse, CH-8910 Affoltern-am-Albis, Switzerland; wolfgang.mayer@medena.ch (W.M.); chiara.deluca@medena.ch (C.D.L.); 4R&D Department, Swiss DEKOTRA GmbH, 549 Badenerstrasse, CH-8048 Zurich, Switzerland; trakhtman@dekotra.com; 5Centre of Innovative Biotechnological Investigations Nanolab (CIBI-NANOLAB), 197 Vernadskiy Pr., 119571 Moscow, Russia

**Keywords:** heavy isotopes depleted water, hormesis, redox status, pro-/anti-inflammatory cytokines, recovery time, glucose metabolism, energy metabolism

## Abstract

Water depleted of heavy isotopes, such as ^2^H_1_ and ^18^O_16_ (HIDW), has shown numerous biological/health effects *in vitro*, *in vivo*, and in epidemiological studies. Major observations were related to cell growth/differentiation, immune/nervous system responses, endurance/adaptation, mitochondrial electron transfer, energy production, glucose metabolism, etc. No human studies to confirm physiological, metabolic, and immune responses to the consumption of HIDW have been performed. A placebo-controlled study on healthy volunteers (n = 50) under fitness load who consumed 1.5 L HIDW (58 ppm ^2^H and 1780 ppm ^18^O) or normal water for 60 days was carried out. Plasma content of ^2^H_1_ and ^18^O_16_, markers of energy, lipid, and glucose metabolism, anthropometric, cardio-vascular, oxidant/antioxidant, and immunological parameters were determined. Significant decrease in plasma heavy isotopes in the group consuming HIDW was observed in concomitance with an increase in ATP, insulin, and LDH, and diminished plasma lactate. Several anthropometric and cardio-vascular parameters were improved as compared to placebo group. Lipid markers demonstrated antiatherogenic effects, while oxidant/antioxidant parameters revealed HIDW-induced hormesis. Antibacterial/antiviral immunity was remarkably higher in HIDW *versus* placebo group. **Conclusions**: HIDW consumption by humans under fitness load could be a valid approach to improve their adaptation/recovery through several mechanisms.

## 1. Introduction

Being the largest constituent of all living organisms, water is an essential requirement for life. Intra- and extra-cellular water plays numerous physiological roles, providing an appropriate medium for biochemical reactions in cells, cell-to-cell interactions, and cellular responses to biotic and abiotic stresses [1].

The water molecule consists of hydrogen and oxygen, both these elements having more than one stable naturally-occurring isotope. The most abundant hydrogen isotope in surface water has an atomic mass 1 (^1^H_1_, protium), while the isotope with atomic mass 2 (^2^H_1_, deuterium) is present in small quantities (156 ppm or 0.0156% of all hydrogen atoms) [2]. The ratio of the two stable hydrogen isotopes strongly influences the chemical and physical properties of water [3,4]. The most abundant natural oxygen isotope has a mass number of 16 (^16^O_8_), the ^18^O_8_ isotope is present at about 0.2%, and there is also a tiny amount of ^17^O_8_.

The average concentration of natural water containing deuterium is approximately 330 mg/L (17.3 mM), while the concentration of water containing ^18^O_8_ is about 2 g/L (100 mM) [5].

As compared to other biologically important microelements and organic substances in the human body, such as Ca^2+^, Mg^2+^, K^+^ and glucose, the normal ranges of blood serum concentrations of hydrogen and oxygen isotopes are quite high: Ca^2+^ (2.24–2.74 mM); Mg^2+^ (0.75–1.20 mM); K^+^ (3.5–5.1 mM); glucose (3.3–6.1 mM); ^2^H_1_ (12–14 mM); and ^18^O_8_ (100–110 mM) [6]. Previous studies suggested that the hydrogen isotopic composition of human body biomolecules strongly reflects their content in natural water/food, as well as the isotopic fractionation during the biochemical redox reactions [7,8].

It is known that relatively small fluctuations in deuterium content cause changes in the physical-chemical properties of intracellular water [3,4] and have a marked impact on the dynamics of biochemical, cellular, tissue, and systemic regulatory processes [9,10]. 

Studies have clearly shown the antitoxic effect of deuterium-depleted water [11], as well as its positive effects on the state of various protective/adaptation systems of the body [10,12], the activity of which declines with age [13].

From recent publications, it has become clear that deuterium- and ^18^O_8_-depleted light water substantially improve the functions of mitochondria [14]. These essential organelles are producers of energy in the unique form of adenosine-3-phosphate (ATP), a chemical storage of energy being supplied upon request to all energy-requiring biological processes. This implies potential benefits to mitochondria-rich tissues, such as muscles and nerves. From a mechanistic point of view, light water stimulates signalling for insulin receptor in the classical insulin target tissues, such as fat, muscle, and the liver, where insulin is essential for regulating energy-producing glucose and lipid metabolism [15].

Recently, it has become widely recognised that exercise and regular physical activity exert a number of positive effects on the human organism by decreasing adiposity and mitochondrial dysfunction, by balancing redox status and immune functions, and by enhancing anti-inflammatory and metabolic capacities [16]. Since reactive oxygen and nitrogen species (RONS) regulate insulin signalling to cells—being its second messengers—and also exert insulin-like effects (reviewed in [17]), possible alterations in the RONS-insulin axis upon consumption of heavy isotopes-depleted water would definitely affect insulin-dependent energy metabolism during moderate physical exercise. 

These observations from basic research gave a strong impulse to evaluate the effects of drinkable water depleted from hydrogen and oxygen heavy isotopes on healthy human beings subjected to regular physical training, in terms of energy expenditure and recovery, as well as of essential physiology parameters, and markers of individual physical capacity.

Goal of the present human study was the evaluation of the effects of consumption of water depleted from heavy isotopes of hydrogen and oxygen (58 ppm ^2^H_1_ and 1780 ppm ^18^O_8_, respectively) on physical recovery, anthropometric characteristics, redox parameters, metabolic and immune markers of healthy volunteers under regular moderate physical load in a fitness centre.

## 2. Materials and Methods

### 2.1. Study Design

The study protocol was scrutinised and approved by the local Ethical Committee of the Berbekov’s State Medical University (Protocol 53/2-2017 of 23 January 2017). Informed consent was obtained from all participants prior to enrolment and data collection. Collected data included: personal and anamnestic records, blood sampling for the specific sets of analyses, and the storage of deep-frozen blood fractions in a blood bank, in accord with Helsinki Declaration on ethics in human experimentations.

The tested cohort consisted of 50 healthy adults of both sexes aged from 20 to 46 years, randomly placed into two groups. Volunteers with overt endocrine pathology, acute illnesses, heart diseases, uncontrolled hypertension, ongoing use of hypnotics, or any treatment for breathing disorders were excluded from enrolment.

**The Control group** (CTR, n = 25, mean age 28.6 ± 10.4 years, age range 20–44 years) of volunteers was under regular moderate physical load at fitness centre (1 h/day for 5 consecutive days a week). The participants consumed natural water of local sources (tap or spring water) *ad libitum* in accordance with individual habits. The intensity of training process was routinely controlled by conventional computer programs (Polar, Garmin, and others). **The Experimental group** (EXP, n = 25, mean age 28.2 ± 8.8 years, age range 21–46 years) volunteers were under similar physical load at fitness centre (1 h/day for 5 consecutive days a week). The participants consumed 1.5 L a day of water depleted of the heavy isotopes ^2^H_1_ and ^18^O_16_ (HIDW). The administered HIDW had a content of heavy-water isotopes of 58 ppm ^2^H and 1780 ppm ^18^O (see Section 2.2). 

The duration of the testing was 2 months. The HIDW administered to the experimental group, produced by the company Langway L.t.d., Moscow, was duly certified as drinking water for humans and distributed under the corresponding Russian Federation law.

### 2.2. Drinking Water under Investigation

The HIDW under study was manufactured as follows. The water originated from Caucasus Mountain glacier sources of the Kabardino-Balkar Republic, Russian Federation, was depleted from heavy hydrogen and oxygen isotopes by a patented column chromatographic method and then remineralised with dehydrated salts. The isotopic content, mineralisation, and physical parameters of HIDW are collected in the Table 1 and Table 2. The HIDW drinking water was bottled and distributed to participants of the experimental group of volunteers. 

The control group participants were recommended to drink tap or glacier spring water they were used to. The local tap and glacier waters were analysed for their isotopic content and results are present in Table 1 to compare them with those of HIDW.

### 2.3. Assessment of Anthropometric Changes

Anthropometric data were registered by trained medical personnel twice, in the beginning and at the cessation of the trial. Ethnicity was self-reported and captured for descriptive purposes.

As anthropometric markers, body weight, body height and body mass index (BMI) were measured. In addition, chest, waist, hips, thigh, and forearm circumferences were measured in the beginning and at the cessation of the study. 

Standing height was measured to the nearest 0.1  cm (without shoes and socks) using a wall-mounted human growth meter. Measurements were repeated, and a third measurement was taken if the first two differed by more than  0.3  cm. The two closest measurements were averaged. Weight was measured on the naked subjects to the nearest 0.1  kg using a scale. Three measurements were taken if the first two measurements differed by more than 0.5  kg. The two closest measurements were averaged. 

BMI was calculated as a ratio of body weight in kg to square height in m^2^ (kg/m^2^). BMI-determined weight categories were determined according to the World Health Organisation recommendations [18]: normal or healthy BMI range (18.5–24.9  kg/m^2^), overweight (25.0–29.9  kg/m^2^), or obese (≥30  kg/m^2^).

Chest, waist, hips, thigh, and forearm circumferences were measured using a nonelastic anthropometric measuring tape to the nearest 0.1  cm. Two measurements were taken, with a third required only if the first two differed by > 0.5  cm. The two closest measurements were averaged.

### 2.4. Assessment of Heart/Vascular Physiology Parameters

The cardiac/vascular physiology markers were registered 3 times, at the trial entrance (time point 0 days, t0), after 1 month (time point 30 days, t30), and at the cessation of the trial (time point 60 days, t60).

The Ruffier–Dickson Index (RDI) (controlled knee-bending as a physical load) [19] was applied to quantify patient’s background fitness and recovery after moderate physical load. The RDI was calculated by the formula:4 × (P1 + P2 + P3) − 200) : 10,(1)
where P1—normal pulse rate; P2—pulse rate immediately after 30 squats; P3—pulse rate 1 min later 30 squats. All pulse rates were measured for 15 s. Physiological meaning of the RDI values: 0–3—very good physical state (perfect fitness); 3–5.9—good physical state (good fitness); 6–10.9—sufficient physical state (sufficient fitness); 11 and more—insufficient physical state (bad fitness).

The Orthostatic Index (OI) (movement from a lay down to standing position) was measured to determine both baseline and physical load-dependent status of the patients. The OI was calculated by the formula: mPR (laying) − mPR (standing),(2)
where mPR (laying) is a mean value of pulse rates measured 4 times during 1 min in laying position, and mPR (standing) is a mean value of pulse rates measured during 1 min in subsequent standing position. Physiological (fitness related) meaning of OI values: 0–12 beats/min—good training/fitness; 13–18 beats/min—sufficient fitness; 18–25 beats/min—no fitness; >25 beats/min—overload or cardio-vascular insufficiency/disease or other kinds of disease.

Electrocardiography measurements were carried out in the beginning (t0) and after the cessation of the trial (t60). The results were analysed by a cardiologist, and their physiological/pathological meanings were described in detail.

### 2.5. Reagents and Assay Kits

The majority of chemical reagents, HPLC standards, mediums, solvents, and luciferin-luciferase for ATP assay were from Sigma Chemical Co. (St. Louis, MO, USA); kits for enzyme activity assays, and Griess reagent for nitrites/nitrates determination were from Cayman Chem. Co. (Ann Arbor, MI, USA). Manufacturers of other reagents are mentioned within the respective methods.

### 2.6. Assessment of Heavy Water Isotopes (^2^H and ^18^O) in Different Water Samples and in Human Plasma

The deuterium concentration in water samples and the blood plasma was determined by a nuclear magnetic resonance (NMR) spectrometer (JEOL JNM-ECA 400 MHz, Tokyo, Japan). Spectrograms were made at the corresponding resonant frequency of deuterium nucleus—61.4 MHz. The spectrogram parameters are as follows: 6.7 s (acquisition time), 20 s (relaxation delay), 5.6 µs (x-pulse), and 0.15 Hz (resolution). Spectrogram temperature—25 °C, with the stabilisation accuracy of 0.2 °C. Measurements were performed using ampoules 5 mm in diameter, inside of which sealed capillaries were rigidly fixed. The latter contained mixtures of deuterated and nondeuterated dimethyl sulfoxide (DMSO), calibrated in the concentration scale under determination. This yielded a ^2^D (deuterium) NMR signal in the region of 3.4 ppm (with respect to (CD_3_)_4_Si), while the ^2^D NMR signal of HDO lay in the field of 4.7 ppm (with respect to (CD_3_)_4_Si). Processing of obtained spectra was aimed at determining the integral intensities of a ^2^D HDO NMR signal, contained in the sample, relatively to the ^2^D DMSO-D_1_ NMR signal, the intensity of which, in turn, was determined under the same conditions in terms of norm types—water samples with a strictly definite deuterium content (3.7, 51, and 150 ppm). Measurements of each sample were repeated 6 times. The accuracy of the deuterium content determination in water and blood plasma samples was ±2 ppm [20]. Measurements calibration for liquids was performed relatively to international standard mean ocean water (SMOW) values (δ^2^H = 0‰; δ^18^O = 0‰) and water corresponding to internal standards.

### 2.7. Redox and Oxidation Marker Assays 

Complete differential blood cell counts and metabolic analyses were performed on fresh ethylene diamine tetra-acetic acid (EDTA)-anti-coagulated peripheral blood of 12 h-fasting subjects. Biochemical assays were performed on peripheral blood plasma, red blood cells (RBC), or total white blood cells (WBC) either immediately (ATP), or within 72 h, on sample aliquots stored at −80 °C under argon. Plasma levels of nitrites/nitrates (NO_2_^−^/NO_3_^−^, expressed as μM) were measured spectrophotometrically by Griess reagent [21]. Protein content was measured by Bradford method [22] using a microplate assay kit (Bio-Rad, Hercules, CA, USA). Reduced glutathione quantitation was conducted by spectrophotometric method as described elsewhere [23]. Plasmatic Cu, Zn superoxide dismutase 3 (Cu,Zn-SOD3, U/g protein) activity was measured spectrophotometrically at 505 nm using kits from Cayman Chemical Company (Ann Arbor, MI, USA) [24,25]. RBC were lysed in hypotonic solution, and the post-spin cell lysates were analysed. Total RBC glutathione S-transferase (GST, U/mg Hb) activity was measured spectrophotometrically by the methods described previously, using chloro-2,3-dinitrobenzene as substrate [26]. RBC glutathione peroxidase (GPX, U/g Hb) activity was determined using Cayman Chemical kit, according to the method [27]. Spontaneous superoxide production by WBC was recorded by lucigenin-amplified chemiluminescence as described elsewhere [28].

### 2.8. ATP Assay

In total, 100 μL of either RBC or WBC pellet was stored on ice until analysis. Ice-cold water (990 μL) was added to 10 μL of cell pellet, and mixed and the lysed cells were kept on ice. The principle of ATP assay is based on the quantitative bioluminescent determination of adenosine 5’-12 triphosphate (ATP), assessed by the Bioluminescence Assay Kit. In the assay, ATP is consumed when firefly luciferase catalyses the oxidation of D-luciferin to adenyl-luciferin which, in the presence of oxygen, is converted to oxyluciferin with light emission. This second reaction is essentially irreversible. When ATP is the limiting reagent, the light emitted is proportional to the ATP present. The measurements of luciferin-luciferase chemiluminescence were performed on a Victor2 1420 multilabel counter, equipped with Wallac 1420 Software (Perkin Elmer, Boston, MA, USA). Results were expressed as mM [29].

### 2.9. Biochemical Analyses

Routine clinical biochemical analyses included: total blood count, detailed plasma lipid profile (total cholesterol (CH) level, the ratios CH/high density lipoproteins (HDL, CH/low density lipoproteins (LDL), CH/very low density lipoproteins (VLDL), plasma glucose and insulin levels, plasma C-reactive protein, adiponectin, and lactate contents, as well as lactate dehydrogenase activity. On the basis of the results obtained, the insulin resistance (IR) Index, the atherogenic index, and the HOMA (homeostatic model assessment) index were calculated by standard formulas. The equation for IR-HOMA calculation:IR-HOMA = Glu (mM) × Ins (mU/L) / 22.5(3)
was used with an assumption that all individuals were healthy, with age range 30–50 years, normal body weight, and 100% of functioning β-cells.

### 2.10. Immunological Assays

The capacity of the WBC to phagocyte bacteria was determined by the following methods: luminol-dependent chemiluminescence activated by opsonised bacteria [28], phagocytosis index (number of bacteria engulfed by a single phagocyte), phagocyte number (number of active phagocytes), and intracellular killing of engulfed bacteria. 

The plasma levels of interleukin 1β (IL-1β,) interleukin 6 (IL-6), interferon gamma (IFN-γ), tumour necrosis factor alpha (TNF-α), and interleukin 10 (IL-10) were measured by the enzyme-linked immunosorbent assay (ELISA) using appropriate antibodies purchased from R&D Systems (Minneapolis, MN, USA), following manufacturer’s instructions. Cytokine concentrations were expressed in pg/mL of plasma. Each protein was quantified in the linear range of its calibration curve.

### 2.11. Statistical Analysis

Statistical analysis of clinical data was carried out using WINSTAT programs for personal computers (Statistics for Windows 2007, Microsoft, Redmond, WA, USA). All biochemical and molecular measurements were performed in triplicate, and data were statistically evaluated. Values were presented as mean, standard error of the mean, 1.96× standard error of the values of triplicate analyses. When the two datasets were compared, data were analysed by Student’s *t* test for unpaired data. Differences between initial/final data for a single participant were analysed by paired *t* test and by Mann–Whitney test for changes from baseline. All reported *p* values are from two-tailed tests, and *p* values of less than 0.05 were considered to indicate statistical significance.

## 3. Results

### 3.1. Effects of HIDW Consumption on Plasma Content of ^2^H and ^18^O Isotopes

The plasma levels of heavy water isotopes such as deuterium (^2^H, D) and oxygen-18 (^18^O) were significantly diminished in the EXP group subjects consuming HIDW (1.5 L a day for 60 consecutive days), while their levels remained unchanged in the CTR group consuming either local tap or spring water (Table 3). A statistically significant difference between the two groups was reached by the 60th day. The baseline distribution of ^2^H and ^18^O plasma levels in the CTR and EXP groups is shown in Figure A1 (Appendix A).

### 3.2. Effects of HIDW Consumption on Anthropometric Parameters

The body weight controlled three times in both test groups was statistically significantly (*p* < 0.01) diminished in the EXP group, while its average value did not change in the CTR group of volunteers (Table 4). The planned measurements of the chest, waist, hip, thigh, and forearm circumferences were repeated three times: in the beginning, in the middle, and upon the cessation of the trial. These values were statistically significantly decreased only in the EXP group.

### 3.3. Effects of HIDW Consumption on Aerobic Endurance, Fitness Levels, and Recovery Rate

The initial BMI values of all participants were within the range of age-related normality. Only two participants (one from the CTR and another one from the EXP group) had BMI = 30, which corresponded to slight (grade I) obesity. The data on BMI calculated for both groups clearly demonstrated that the index had remained unchanged (Appendix B. Table A1). 

Among the physiological parameters determined under moderate physical load, there were the Ruffier–Dickson index and the orthostatic index. These indexes reflect the adaptive capacity of the organism, first of all of the cardiovascular system response, to a controlled and moderate physical load. In addition, the same indexes correlate with the potential for recovery from physical fatigue load. The results of measurements and the physiological meanings of the values obtained are collected in Table 5 and Table 6, respectively. The background mean values of the RDI for the participants of both groups were within the range of sufficient fitness state, in accordance with the physiological meaning graduation. While the RDI did not significantly change in the CTR group of patients, it was substantially and gradually improved (diminished) in the EXP group. Statistical significance of *p* < 0.05 (day 0 versus day 60) was reached after 2 months of experimental water drinking.

The initial mean values of OI were within good physical state/fitness for both EXP and CTR groups (Table 6). During the trial, the mean value slightly fluctuated in the CTR group although never reaching a significant difference with the values registered at the clinical study start. On the other hand, the mean value of the OI in the EXP group improved significantly already after 1 month (*p* < 0.05). Statistical significance was further increased by the end of the trial (*p* < 0.01).

Electrocardiography was recorded and described by an experienced doctor-cardiologist in the beginning and at the cessation of the clinical study (scans of electrocardiography records are collected in the Appendix C. (Scan A1, Scan A2, Scan A3, and Scan A4). The doctor found that, before entering the trial, a major part of the volunteers (n = 41) did not show any abnormalities in the cardiography parameters. However, four people from the CTR group and five people from the EXP group exhibited initial stages of metabolic abnormalities in cardiac muscles, as revealed by characteristic changes in the electrocardiography peaks (V4–V6). When analysed for the second time, none of the volunteers of the EXP group showed negative metabolic changes of electrocardiography curves (Appendix C. ScansA1–A3). At the same time, all four participants of the CTR group with metabolic abnormalities retained the same after the trial (Appendix C. Scan A4).

Obviously, the human cohort was too small to come to definite conclusions that HIDW consumption for at least 2 months could alleviate potentially dangerous baseline metabolic alterations in the myocardium. However, keeping in mind that the improvement of cardiac metabolism corresponded to improved heart functions, HIDW consumption under the conditions of this study could result in a markedly faster recovery and in a better physiological state after moderate physical load (Table 5 and Table 6).

Physiological meaning of RDI values: 0–3.0—perfect fitness; 3.0–5.9—good fitness; 6.0–10.9—sufficient fitness; <10.9—bad fitness.

Physiological meaning of OI values: 0–12 beats/min—good fitness; 13–18 beats/min—sufficient fitness; 18–25 beats/min—no fitness; <25—overload or cardio-vascular insufficiency/pathology or other kind of disease.

### 3.4. Parameters of Energy Metabolism before and after the Trial

In the present study we monitored ATP content in circulating erythrocytes and granulocytes, plasma lactic acid levels, activity of lactate dehydrogenase (LDH), and glucose and insulin levels in plasma. The data of the trial showed that ATP storages in circulating blood erythrocytes, and granulocytes were markedly (*p* < 0.01) increased in the EXP group of people (Figure 1a,b), while ATP levels remained practically unchanged in the CTR group. As was expected from our working hypothesis, LDH activity was found to be increased (*p* < 0.05), and lactic acid plasma content dramatically decreased (*p* < 0.001) in the EXP group only (Figure 1c,d). Based on these facts, one could conclude that the intake of EXP water was favourable to the energy production/storage, likely via an LDH-dependent mechanism.

### 3.5. Effects of Heavy Isotopes Depleted Water Consumption on Lipid and Glucose Metabolism Markers

Our findings of increased energy production/storage decreased body weight and circumferences in participants of the EXP group prompted us to scrutinise lipid and glucose metabolism to reveal possible mechanisms through which HIDW consumption could have such favourable effects. Out of seven lipid metabolism markers studied, i.e., triglycerides, total cholesterol (CH), high density lipoproteins/CH (anti atherogenic marker, HDL/CH), low-density lipoproteins/CH (pro-atherogenic marker, LDL/CH), very low density lipoproteins/CH (pro-atherogenic marker, VLDL/CH), atherogenic coefficient calculated from the major lipid markers/risk factors for atherosclerosis, and adiponectin, a chemokine for adipose cells, only three were significantly and positively changed, exclusively in the plasma of EXP group volunteers. In fact, the anti-atherogenic ratio of HDL/CH was increased (*p* < 0.05) (Figure 2a), the pro-atherogenic ratio of VLDL/CH was decreased (*p* < 0.05) (Figure 2b), and the atherogenic coefficient was diminished (Figure 2c). By the time of second analyses, the mean value of atherogenic coefficient of the EXP group entered into the normality range. There were no significant changes in the lipid pattern of people from the control group (these data are collected in the Appendix B, Table A2).

Concerning glucose metabolism, the following parameters were analysed: fasting glucose and insulin levels, the IR, and the HOMA. The background plasma levels of fasting glucose were within normal range of values for all participants recruited to the study. They were not changed either in the EXP or in the CTR groups. Both IR and HOMA factors were normal in the beginning of the trial. Like glucose, they were not much changed at the trial cessation (Appendix B, Table A2). The levels of fasting insulin in plasma of EXP group were statistically significantly increased (*p* < 0.01) (Figure 2d) although they remained within the normal range of values.

### 3.6. Effects of Heavy Isotopes Depleted Water Consumption on Free Radical Production and Antioxidant Enzymes

Several essential markers of redox imbalance in blood components were detected twice, at the beginning and the end of the clinical study, i.e., superoxide production by circulating WBC, nitric oxide metabolites (nitrites and nitrates in blood plasma), reduced glutathione in RBC, extracellular Cu-Zn superoxide dismutase (EC SOD or SOD3), catalase, and glutathione peroxidase activities in RBC. Only three markers of oxidative stress were significantly and gradually changed after intake of HIDW: the intensity of spontaneous lucigenin—dependent chemiluminescence, a measure of superoxide production by WBC (Figure 3a), the activity of catalase, an enzyme neutralising hydrogen peroxide and organic peroxides (Figure 3b), and the activity of glutathione peroxidase, an enzyme inactivating organic peroxides (Figure 3c). These markers resulted to be significantly increased in the EXP group by the end of the trial, although the mean values remained within the normality range. All the other parameters measured, such as reduced glutathione levels, plasma nitrites/nitrates levels, and activity of SOD3 remained changed (Appendix B, Table A2).

### 3.7. Effects of Heavy Isotopes Depleted Water Consumption on Immune Defence of Circulating White Blood Cells

Three major cellular phagocytosis markers (number of bacteria engulfed by a single white blood cell, phagocytosis index, and intracellular microbial killing) were determined and quantified by the light microscopy. All these markers were significantly increased by the end of the trial in the EXP group while they remained unchanged in the CTR group (Figure 4a–c). Another confirmation of increased intracellular microbial killing in the EXP group was received by measurements of luminol-dependent chemiluminescence in circulating blood phagocytes stimulated by opsonised bacteria (Figure 4d). Collectively, these data clearly show that consumption of heavy isotopes depleted water strongly induced cell-based antibacterial immunity.

### 3.8. Effects of Heavy Isotopes Depleted Water Consumption on Antibacterial and Antiviral Immunity of Circulating White Blood Cells

Practically always, an increased antibacterial response correlates with inflammatory reaction in the organism. Plasma levels of pro- and anti-inflammatory cytokines, such as IL-1β, IL-10, TNF-α, and IFN-γ, as well as C-reactive protein were quantified. A gradually increased content of anti-inflammatory cytokine IL10 and antiviral cytokine IFN-γ were observed exclusively in the EXP group (Figure 5a,b). The plasma levels of C-reactive protein, IL-1β, and TNF-α remained unchanged in both groups studied (Appendix B, Table A2).

## 4. Discussion

In the present trial, the consumption of HIDW for 2 months resulted in significantly diminished plasma levels of both water heavy isotopes as compared to the control group consuming natural tap or spring water (Table 3), while the relative distribution of ^2^H_1_ and ^18^O_16_ remained similar to that of the control group (Appendix A, Figure A1). Similar observations reported in previous publications [30] prompted the authors to draw the conclusion that the isotopic content of plasma depended mainly on that of the drinking water ingested. At the same time, animal experiments have shown that the drop in the plasma levels of heavy water isotopes was not as dramatic as their decrease in several organs, such as pancreas, liver, [30], muscles [31], and kidney [20]. On these grounds, we assumed that the levels of both ^2^H and ^18^O could be significantly decreased not only in plasma but also in the internal organs involved into definite lipid, glucose, and energy metabolic processes.

The body weight decrease in the experimental group was accompanied by diminished circumferences (Table 4) that could be a consequence of at least three losses: (1) of body interstitial water, (2) of lean mass, (3) and/or of fat. Since by technical reasons we did not distinguish lean versus fat mass loss, we were unable so far discuss possible effects of experimental water consumption on these important components of total body weight. It has been shown in previous animal experiments that water depleted from ^2^H_1_ (and simultaneously from ^18^O_16_) possessed remarkable diuretic effect [20]. Therefore, we suggested that the changes in the anthropometric parameters observed in the EXP group could be attributed to enhanced water loss through urination and/or sweating. Recently, it has been suggested [32] that HIDW could be used as a treatment of diet-induced obesity of experimental animals.

The fact that BMI was not changed in both groups (Appendix B, Table A1) did not surprise us because the index reflects body mass (kg) related to body surface (cm^2^). In the case of EXP group, both components of the ratio body mass/body surface were significantly decreased (Table 4), thus the calculated mean BMI values remained unchanged. In the case of the CTR group, BMI did not change since both components of the ratio body mass/body surface remained unchanged.

The positive physiological effects (assessed by RDI and OI tests) were observed in the EXP group (Table 5 and Table 6). The RDI data correlated with the maximal oxygen uptake (VO_2max_) that is considered the best index of cardiorespiratory fitness [33].

The improvement of cardiovascular system metabolism and function upon consumption of HIDW by animals has been previously shown [34]. This corresponds to our observations in the human cohort (Appendix C) and could be at least partly explained by the increase in ATP levels available for cardiac muscle contraction-relaxation, by an improvement of energy-producing metabolism and by a water loss through enhanced urination or sweating. Of course, these limited data of electrocardiography should be considered as preliminary observations. Larger studies may confirm or reject them; therefore, they are not included in the study conclusions.

The screening of biochemical parameters before and after the trial revealed the absence of statistically significant differences between the groups as well as between the start and cessation trial points, for the majority of markers studied (Appendix B, Table A2).

However, three markers of energy metabolism were substantially changed in the EXP group: LDH, ATP, and lactic acid (Figure 1). The interplay of lactic acid, ATP content, and LDH activity in muscular fatigue and tissue acidosis has been a target for multiple molecular, cellular, physiological, and clinical research over last 50 years. The onset of acidosis during physical exercise is commonly attributed to the accumulation of lactic acid, which is thought to be a causative reason for muscle fatigue [35,36]. On contrast, LDH prevents muscular fatigue and possible failure through multiple pathways. For example, NAD+ formed as a byproduct during LDH-catalysed lactate-directed reaction maintains the cytosolic redox potential and promotes ATP generation, thus providing more energy to contracting muscles [37]. Simultaneously, lactic acid as a muscular fatigue-inducing/maintaining factor was found in lower quantities in the EXP group than in the CTR group loaded with the same training program (Figure 1d).

In our opinion, the observed slight insulinemia (Figure 2d), which did not correspond to pathological IR, is an enhanced adaptation response of human organism to moderate physical load caused by the depletion of heavy water isotopes. This physiologically important metabolic adaptation could inhibit glucose uptake into insulin-responsive cells, such as muscles and adipose cells, in order to protect them from the oxidative stress associated with physical exercise [38]. Insulin has even been regarded as a regulatory antioxidant preventing a “lipotoxic stress” under certain conditions including physical exercise [38]. Injections of physiological doses of insulin to healthy humans [39] resulted in remarkably increasing ATP production in *vastus lateralis* muscle mitochondria that was associated with stimulation of oxidative phosphorylation and up-regulation of mitochondrial protein synthesis at both transcriptional and translational levels.

We assume that the major molecular target(s) of HIDW could be mitochondrial chain of electron transport that could cause an enhanced ATP production by mitochondria. Similar data have been reported previously in animal experiments [40]. An accelerated electron transport through mitochondrial respiratory chain is strongly associated with elevated ATP production and superoxide leak from mitochondria. Here, we discussed a highly elevated content of ATP in circulating red and white blood cells of people drinking heavy isotopes-depleted water for 2 months. An increased superoxide production by circulating white blood cells was also observed (Figure 3a). Another target could be an insulin receptor initiating intracellular glucose transport. The reversible adaptive inhibition of insulin receptor in the presence of low levels of heavy water isotopes could alter insulin function. This assumption has been already successfully tested in the *in vitro* and animal experiments [41].

During exercise, the rate of metabolism in skeletal and cardiac muscles increases. The physiological response consists of enhanced oxygen uptake, increased body temperature, and accelerated blood flow. These exercise-connected physiological changes lead to moderately greater-than-baseline RONS production in skeletal muscles [42,43,44,45,46] and in other cells and organs [16,46,47]. Sources other than the electron transport chain enzymes in the mitochondria, such as xanthine oxidase [45,48] and NADPH oxidase [49], contribute to RONS generation during exercise. This transient increase in RONS levels activates redox-responsive nuclear factors NF-κB and AP-1 followed by activation of inflammatory pathways [47] and by over-expression of major antioxidant enzymes, such as superoxide dismutase, glutathione peroxidase, and catalase [16,50]. Of note, exercise-induced superoxide dismutase over-expression in rat skeletal muscles was isoform- and time-dependent: SOD protein levels were increased for maximum 48 h [51].

At low levels, oxidative stress up-regulates certain antioxidant enzymes, such as catalase [16], superoxide dismutase [45], and glutathione peroxidase [52] and boosts the immune response [47]. When intensity of RONS production exceeds the capacity of antioxidant enzymes and nonenzymatic antioxidants to inactivate them, numerous damages to biologically essential molecules (DNA, lipids, proteins, and polysaccharides) take place [53]. Exercise-related oxidative stress also leads to muscle fatigue, reduced athletic performance, and impaired recovery [16,52].

Several essential markers of redox imbalance in blood components were determined twice in the beginning and in the end of clinical study (Figure 3 and Appendix B, Table A2). Some of them remained unchanged while the others, such as superoxide anion-radical production, catalase, and glutathione peroxidase activities were significantly increased. These data suggest that diminished levels of heavy water isotopes caused somehow the hormesis state, which is characterised by the moderate enhance of pro-oxidants and an adaptive induction of endogenous antioxidant enzymes. This hormesis-based mechanism is currently considered as a major self-protection against environmental pro-oxidant hazards, physical exercise-induced oxidative stress, age-related oxidative stress, etc. [16,54]. The decrease in energy levels severely affects the cellular functions requiring ATP, such as biosynthesis, receptor, and channel activation, signal transduction, and cell division [55]. On the other hand, effective ATP synthesis through activation of electron transfer in mitochondria and oxidative phosphorylation is tightly connected with electron leakage and mitochondria-induced oxidative stress [16], which is compensated by endogenous protection systems, such as Nrf2-dependent cytoprotective defence [54]. As a positive feedback loop, Nrf2 activation is essential for increased ATP synthesis [56]. This mechanism of protection from mild stresses is initiated by RONS. Therefore, one could predict that the mild induction of their levels would lead to the activation of protective mechanisms (hormesis or mitohormesis) and to the increased energy stores in human organism [57]. This is a molecular basis for practically all positive effects of moderate physical exercises [16]. Similar hormesis-stimulating and ATP restoring strategy has been recently suggested as a promising pharmacological approach to treat complication of diabetes [58], and obesity [59], as well as age-related pathologies in general [60]. However, HIDW protected H_2_O_2_-induced oxidative stress and subsequent cellular damage through phosphatidyl inositol-3-kinase/AKT signalling pathway [61].

Analyses of the effects of HIDW on lipid metabolism showed (Figure 2a–c) that HDL was statistically significantly increased, while VLDL and the atherogenic factor were decreased (*p* < 0.05) in the EXP group by the end of the study. However, even being changed remarkably, these two lipid markers remained within the normal range. In the beginning of the trial, the mean value of the atherogenic coefficient was above the upper level of normality for both groups, probably reflecting the traditional dietary habits of local people who consume daily a fatty meat (beef, mutton, and lamb), use non-vegetable fats for cooking, and rarely if ever eat fish/marine products.

These data suggested that even a short-term drinking of HIDW decreased significantly lipid risk factors for the development of atherosclerosis and metabolic syndrome, while it did not have any effect on lipid risk markers for obesity, such as leptin (Appendix B, Table A2).

The next set of analyses revealed that immune responses (cellular antibacterial Figure 4 and selected cytokine production—INF-gamma and IL10) were positively affected by drinking light water. At the same time, several pro-inflammatory cytokines, such as IL1β and TNFα, as well as C-reactive protein remained unchanged (Appendix B, Table A2). IFN-gamma belongs to a group of inducible cytokines with a broad range of antimicrobial and antiviral activity. It is secreted from numerous immune cells in an early response to infection prior to onset of the generalised immune reaction (recently reviewed in [62]). IFN-gamma targets practically all types of cells and tissues regulating their growth, differentiation, and apoptosis that leads to the modulation of the immune response (reviewed in [63]). Phagocytes activated by IFN-gamma play essential and multiple roles in innate and adaptive immune responses [64], mainly through inducible nitric oxide synthase and its product nitric oxide (NO), which is a key antimicrobial and antiviral agent [65]. With regard to IFN-gamma, anti-inflammatory IL-10 inhibits IFN-gamma production by Th1 lymphocytes as well as IFN-gamma-induced activation of macrophages, and both cytokines cooperate in the suppression of excessive immune responses in the course of pathogen elimination [66].

The present placebo-controlled human study being a pilot limited trial has a number of serious limitations. Further larger randomised trials on fitness groups as well as on professional sportsmen are needed. The primary and secondary outcomes of HIDW consumption should be better preselected basing on the results obtained in this very first study on human cohort. The positive effects on cardiovascular system and atherogenic potential observed here should be thoroughly investigated in order to reveal molecular mechanisms underlying the metabolic responses of physically loaded organisms to changed ratios of heavy and light water isotopes. Mechanisms of HIDW effects on immune system, redox status, and energy metabolism deserve profound studies at both molecular and cellular levels.

## 5. Conclusions

The very first placebo-controlled study on healthy volunteers (n = 50) under fitness load who consumed 1.5 L heavy isotopes depleted water (58 ppm ^2^H and 1780 ppm ^18^O) or normal water for 60 days was carried out. Plasma content of ^2^H_1_ and ^18^O_16_ were substantially decreased in the experimental group by the end of the study. Significant decrease in plasma heavy isotopes in the group consuming heavy isotopes depleted water was observed in concomitance with an increase in ATP, insulin, and LDH, and diminished plasma lactate. Several anthropometrical and cardiovascular parameters were significantly improved as compared to placebo group and the beginning of the study. Lipid markers demonstrated anti-atherogenic effects while oxidant/antioxidant parameters revealed heavy isotopes depleted water-induced hormesis. Antibacterial/antiviral immunity was remarkably higher in the experimental versus placebo group.

Consumption of heavy isotopes depleted water by humans under fitness load could be recommended to improve their adaptation/recovery through several mechanisms.

## Figures and Tables

**Figure 1 sports-09-00110-f001:**
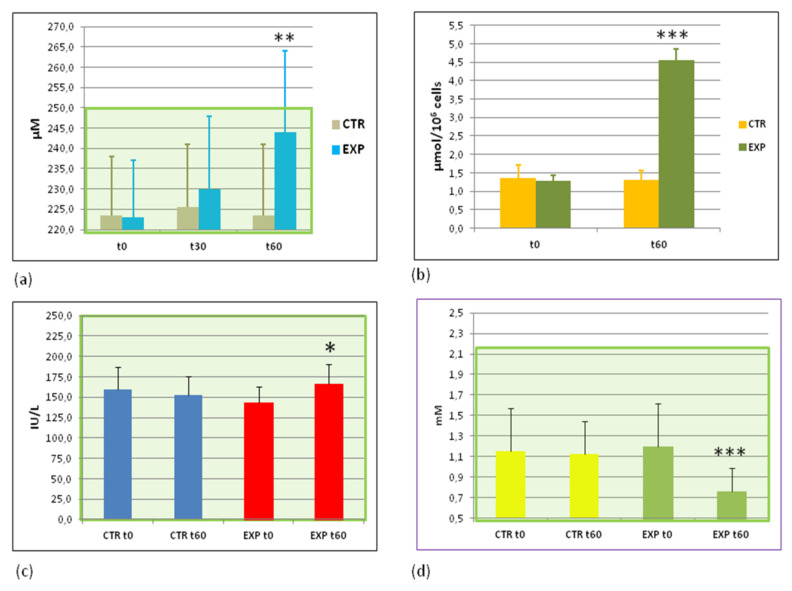
Effects of HIDW consumption on the markers of energy metabolism. (**a**) ATP content in RBC (μM); (**b**) ATP content in WBC (μmol/10^6^ cells); (**c**) lactate dehydrogenase activity in blood plasma (international units/L); and (**d**) lactic acid content in the blood plasma (mM). The green area covers the ranges of normal values. Abbreviations: CTRt0 and CTRt60—the control group before and after the trial, respectively; EXPt0 and EXPt60—the experimental group before and after the trial, respectively. * *p* < 0.05; ** *p* < 0.01; *** *p* < 0.001 vs. EXP t0.

**Figure 2 sports-09-00110-f002:**
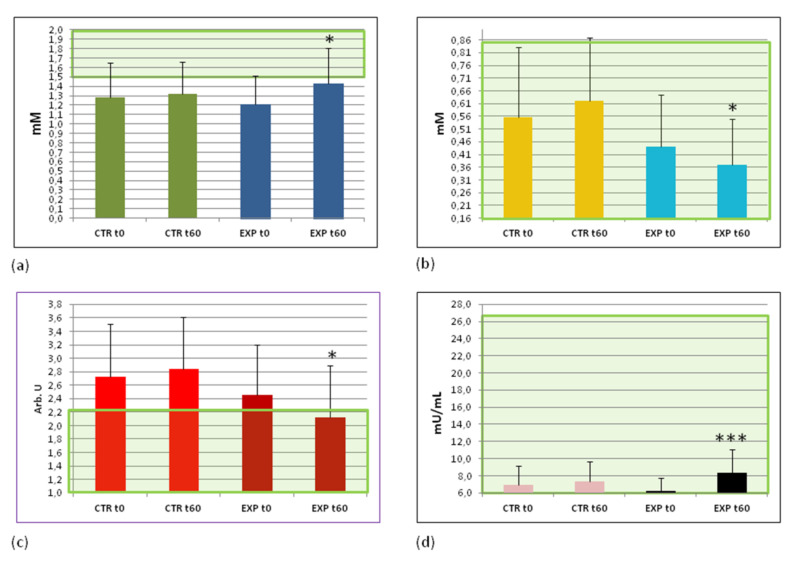
Changes of pro- and anti-atherogenic lipids and insulin levels in the blood plasma (**a**) High-density lipoprotein levels (HDL, mM); (**b**) Very low density lipoprotein levels, VLDL, mM); (**c**) Calculated atherogenic coefficient (arbitrary units); and (**d**) plasma insulin content (mU/mL). The green area covers the ranges of normal values. Abbreviations: CTRt0 and CTRt60—the control group before and after the trial, respectively; EXPt0 and EXPt60—the experimental group before and after the trial, respectively. * *p* < 0.05; *** *p* < 0.001 vs. EXP t0.

**Figure 3 sports-09-00110-f003:**
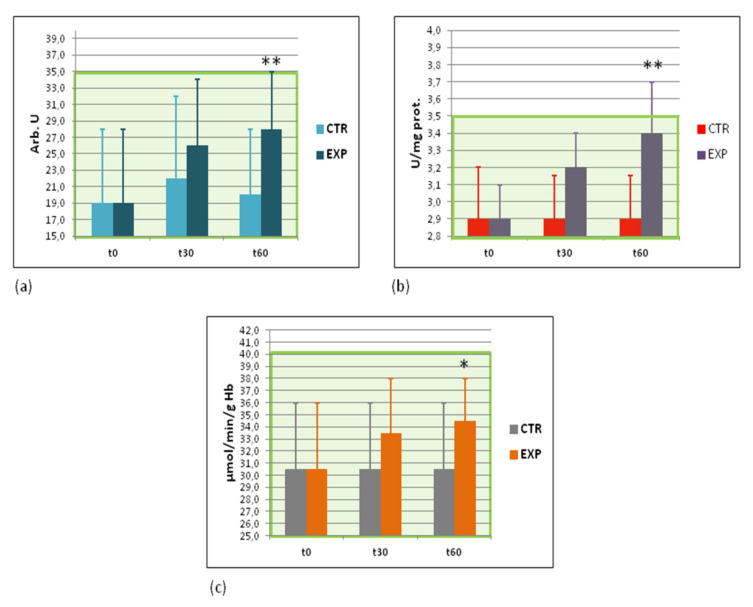
Effects on superoxide anion-radical production by WBC and catalase and glutathione peroxidase activities in RBC. (**a**) Intensity of spontaneous superoxide anion-radical production by white blood cells as assessed by lucigenin-dependent chemiluminescence (arbitrary units); (**b**) activity of RBC catalase (U/mg protein); and (**c**) activity of RBC glutathione peroxidase (μmol/min/g Hb). The green area covers the ranges of normal values. t0—before the study; t30—at the 30th day of the study; t60—at the 60th day, cessation of the study; * *p* < 0.05; ** *p* < 0.01 vs. EXP t0.

**Figure 4 sports-09-00110-f004:**
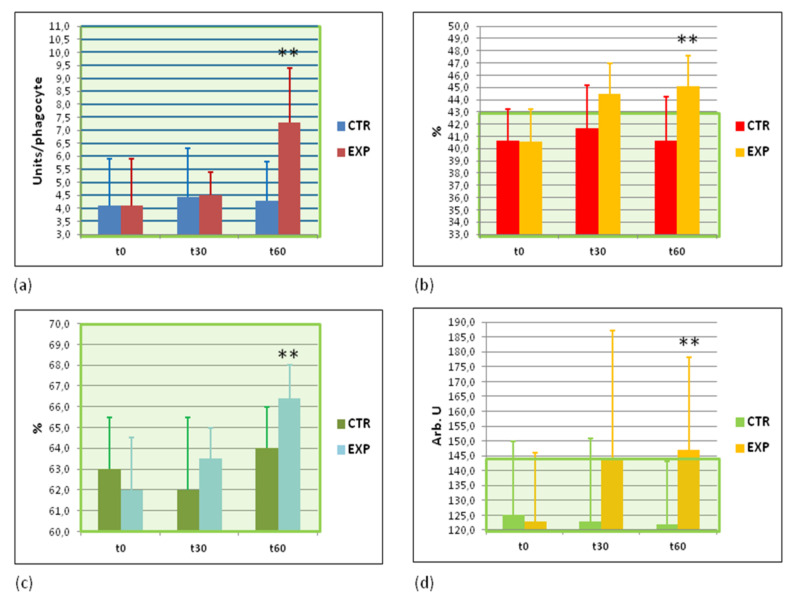
Cellular immunity against bacterial infections is remarkably increased. (**a**) Phagocytosis intensity assessed by a number of bacteria engulfed by single phagocyte (bacterial units per white blood cell); (**b**) percentage of active phagocytes, %; (**c**) intensity of intracellular bacterial killing, %; (**d**) hydroxyl radicals and nitrogen peroxide release from white blood cells as assessed by the intensity of opsonised zymosan—stimulated luminol—dependent chemiluminescence (arb. units, arb. U). Green area corresponds to normal values. CTR—control group; EXP—experimental group. t0—before the study; t30—at the 30th day of the study; t60—at the cessation of the study. ** *p* < 0.01 vs. EXPt0.

**Figure 5 sports-09-00110-f005:**
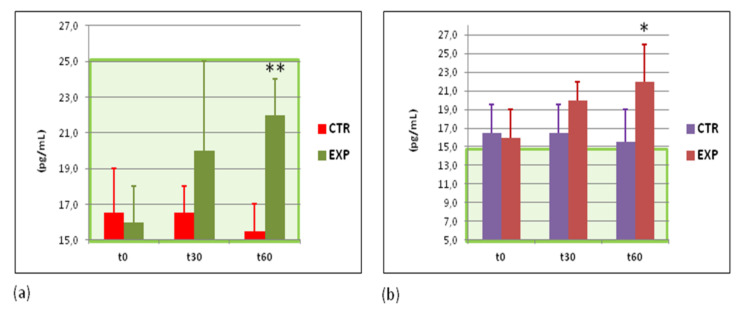
Plasma levels of interferon gamma and interleukin 10 proteins. (**a**) Levels of interferon gamma (IFNγ) protein (pg/mL); and (**b**) levels of anti-inflammatory interleukin 10 (IL-10) protein (pg/mL). Green area corresponds to normal values. CTR—control group; EXP—experimental group. t0—before the study; t30—at the 30th day of the study; t60—at the cessation of the study. * *p* < 0.05; ** *p* < 0.01 vs. EXPt0.

**Table 1 sports-09-00110-t001:** Absolute (ppm) and Relative to Vienna Standard Mean Ocean Water (‰) Isotopes ^2^H and ^18^O content in the experimental (heavy-isotope depleted, HIDW) and in the control natural drinking water (tap and spring).

Water Samples	^2^H Content		^18^O Content	
	ppm	δ^2^H_v-SMOW_ *, ‰	ppm	δ^18^O_v-SMOW_ *, ‰
HIDW	58.5 ± 0.3	−624.4 ± 1.9	1780.7 ± 0.1	−111.9 ± 0.4
Natural local tap water	146.8 ± 0.3	−57.5 ± 1.7	1981.3 ± 0.2	−11.8 ± 0.7
Natural local spring water	148.1 ± 0.3	−49.2 ± 1.9	1984.5 ± 0.2	−10.2 ± 0.8

* δ^2^H_v-SMOW_ and δ^18^O_v-SMOW_ difference relative to Vienna standard mean ocean water (VSMOW).

**Table 2 sports-09-00110-t002:** Mineral content and physical characteristics of heavy-isotope depleted water (HIDW) consumed by the experimental group volunteers.

Mineral/Anion/Physical Parameter	Value, Units
Potassium	4.4 mg/L
Sodium	10.0 mg/L
Calcium	50.1 mg/L
Magnesium	10.4 mg/L
Chloride	70.0 mg/L
Fluoride	0.5 mg/L
Sulphate	10.0 mg/L
Hydrocarbonate	70.8 mg/L
Total mineralisation	226.0 mg/L
Electroconductivity	320 μS/cm
pH	7.2

**Table 3 sports-09-00110-t003:** Consumption of HIDW (1.5 L/day for 60 consecutive days) resulting in the diminished plasma levels of heavy isotopes (^2^H and ^18^O).

Isotope/Group	Plasma ^2^H Content, ppm		Plasma ^18^O Content, ppm	
	Before (t0)	After (t60)	Before (t0)	After (t60)
EXP HIDW (n = 25)	148.5 ± 0.1	129.9 ± 1.0 ***	1986 ± 1.0	1961 ± 10.0 *
CTR Local water (n = 25)	148.5 ± 0.1	148.5 ± 0.1	1990 ± 3.0	1990 ± 1.0

* *p* < 0.05 vs. t0; *** *p* < 0.001 vs. t0.

**Table 4 sports-09-00110-t004:** Changes of anthropometric values after consumption of HIDW (1.5 L/day for 60 consecutive days).

Change of the Parameter (t60–t0), Units	Group		Significance of the Change
	Control	Experimental	
Body weight, kg	0.04 ± 0.03	−0.59 ± 0.32 **	*p* < 0.01
Chest circumference, cm	0.35 ± 0.10	−0.52 ± 0.28 ***	*p* < 0.001
Waist circumference, cm	0.04 ± 0.03	−0.70 ± 0.19 **	*p* < 0.01
Hip circumference, cm	0.09 ± 0.06	−0.63 ± 0.21 **	*p* < 0.01
Thigh circumference, cm	−0.04 ± 0.01	−0.50 ± 0.12 **	*p* < 0.01
Forearm circumference, cm	−0.04 ± 0.01	−0.17 ± 0.05 **	*p* < 0.01

** *p* < 0.01 vs. t0; *** *p* < 0.001 vs. t0.

**Table 5 sports-09-00110-t005:** Physical load Ruffier–Dickson index (RDI) changes expressed in arbitrary units (normal range < 10.9).

CTR			EXP		
t0	t30	t60	t0	t30	t60
9.1 ± 1.3	9.0 ± 1.8	8.9 ± 1.9	10.2 ± 1.5	9.2 ± 1.9	8.0 ± 1.1 *

* *p* < 0.05 vs. t0.

**Table 6 sports-09-00110-t006:** Orthostatic index (OI) changes expressed in arbitrary units.

CTR			EXP		
t0	t30	t60	t0	t30	t60
8.6 ± 0.7	8.2 ± 1.2	7.7 ± 1.9	9.3 ± 1.3	8.1 ± 0.6 *	6.6 ± 0.8 **

* *p* < 0.05 vs. t0; ** *p* < 0.01 vs. t0.

## Data Availability

The archives of raw data obtained in the trial are stored at the Department of Microbiology, Virology, and Immu-nology, Kabardino-Balkar Berbekov’s State University.

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
