# Peer review of "Effects of Heavy Isotopes (2H1 and 18O16) Depleted Water Con-Sumption on Physical Recovery and Metabolic and Immunological Parameters of Healthy Volunteers under Regular Fitness Load"

_sports, 2021, doi:10.3390/sports9080110_

Round 1

Reviewer 1 Report

The article "Effects of heavy isotopes (2H1 and 18O16) depleted water consumption on physical recovery and metabolic and immunological parameters of healthy volunteers under regular fitness load" is written in good scientific language, the study is well planned and performed at a high level, the data are well presented, however there are several comments for consideration.

  1. In the section introduction and discussion, it is necessary to expand the information on the effects of the isotope of H1 in nature living systems. The list of references contains mainly articles from 2000-2010. However, at the same time, over the past few years 2018-2021, the scientific community has significantly expanded knowledge on the effect of different ratios of hydrogen isotopes on living systems in vitro and in vivo. For example, at the cellular level, some pathways and mechanisms are considered that can expand the understanding of the data obtained by the authors. Consideration should be given to adding this information.
  2. In the section, methods should indicate the composition and caloric content of the diet, which was fed to volunteers before and during the experiment because it studies metabolism and some immunological parameters.
  3. A paragraph with limitations of the study in discussion section is needed.
  4. Perhaps the conclusions should be structured in one paragraph without subparagraphs 1, 2, 3 ... Also, in the conclusions, the most important results of the scientific research should be presented more succinctly.

Author Response

REWIER 1

The article "Effects of heavy isotopes (2H1 and 18O16) depleted water consumption on physical recovery and metabolic and immunological parameters of healthy volunteers under regular fitness load" is written in good scientific language, the study is well planned and performed at a high level, the data are well presented, however there are several comments for consideration.

  1. In the section introduction and discussion, it is necessary to expand the information on the effects of the isotope of H1 in nature living systems. The list of references contains mainly articles from 2000-2010. However, at the same time, over the past few years 2018-2021, the scientific community has significantly expanded knowledge on the effect of different ratios of hydrogen isotopes on living systems in vitro and in vivo. For example, at the cellular level, some pathways and mechanisms are considered that can expand the understanding of the data obtained by the authors. Consideration should be given to adding this information.

A1.     We understand and appreciate your concern regarding the absence of expanded knowledge “on the effects of different ratios of hydrogen isotopes on living systems in vitro and in vivo”. We admit that there were many recent publications, mainly, on these effects towards microbial, plant, and human cancer cells. In this manuscript, we tried to limit somehow a number of references on original research including instead recent comprehensive review articles (See references Basov et al., 2019; Gyongyi et al., 2013; Strekalova et. al., 2015; Dzhimak et al., 2014; Pomytkin, 2012) as well as focusing on adaptation, energy-producing or energy-consuming, and insulin-dependent effects of water depleted of heavy isotopes. However, we added two recent publications (ref. 40 and ref. 61 of the years 2019 and 2020, respectively) to the list of references.

The 2nd reviewer recommended us to shorten the list of references.

  1. In the section, methods should indicate the composition and caloric content of the diet, which was fed to volunteers before and during the experiment because it studies metabolism and some immunological parameters.

A2.     The the Methods section, Study design sub-section, the description of composition and caloric content of the diet were added following your indications. It now reads as: “All recruited volunteers were on fitness program of body weight maintenance (no body weight gain or loss). The calories of daily diet during the trial were calculated individually taking into account sex, age, and weight. On average, daily calories consumption for males ranged from 2400 to 2700 kcal while for females from 1900 to 2200kcal. The food contained 30% proteins, 20% fat, and 50% carbohydrates.” (highlighted in yellow colour).

  1. A paragraph with limitations of the study in discussion section is needed.

A3.     The limitations of the study are listed in the end of Discussion section as per your suggestion (highlighted in yellow colour).

  1. Perhaps the conclusions should be structured in one paragraph without subparagraphs 1, 2, 3 ... Also, in the conclusions, the most important results of the scientific research should be presented more succinctly.

A4.     The Conclusions section was edited substantially in accord with your indications.

The authors express their gratitude to the reviewer for the swift reply and very professional and exact comments aiming to improve the quality of our MS.

Reviewer 2 Report

The MS is of a potentially interesting topic. Improving fitness and metabolic capacity by altering the isotopic composition of water is a novel and promising field of research.

The MS has, however, several major flaws that need profound reworking before publication can be decided upon.

A thorough language correction is inevitable, preferably by a professional editing service. The authors were – inevitably – influenced by their native language which made the English text clumsy and hard to follow.

The methods used are adequate. Their description is not always easy to follow.

The presentation of the results is mostly clear.

Discussion is too lengthy and divergent. It should be more focussed on interpretation of the main results. Also, the references are probably too numerous. Non-essential ones could be removed, first of all where 3, 4 or even more papers are cited to the same statement.

I have inserted a lot of particular comments in the text, the file is attached.

Author Response

REVIEWER 2

The MS is of a potentially interesting topic. Improving fitness and metabolic capacity by altering the isotopic composition of water is a novel and promising field of research.

The MS has, however, several major flaws that need profound reworking before publication can be decided upon.

  1. A thorough language correction is inevitable, preferably by a professional editing service. The authors were – inevitably – influenced by their native language which made the English text clumsy and hard to follow.

A1. The text was thoroughly edited by our native English speaking colleague.

2. The methods used are adequate. Their description is not always easy to follow.

A2. Thanking you very much for the appreciation of the methods used, we should admit that sometimes modern isotopic, molecular, and cellular methods are extremely complicated as their standard description in detail is difficult to understand by non an expert in the field. At the same time, they could be useful for researchers who would repeat them.

3. The presentation of the results is mostly clear.

A3. Thank you for this favourable comment

4. Discussion is too lengthy and divergent. It should be more focussed on interpretation of the main results. Also, the references are probably too numerous. Non-essential ones could be removed, first of all where 3, 4 or even more papers are cited to the same statement.

A4. Discussion was edited and shortened as per your suggestions. The citations of 3 or 4 papers to the same statement usually relate to different experimental systems or organs, where effects of water heavy isotopes depletion were observed. We tried to limit the number of references including mainly review articles rather than the original research papers. The reviewer 1 of the MS reproached us for insufficient number of publications on the matter. 

5. I have inserted a lot of particular comments in the text, the file is attached.

A5. We followed all your precious suggestions and edited the text accordingly (highlighted in yellow colour)

The authors appreciate very much your accurate and scrupulous analysis of the MS. They were very helpful to improve the readability and quality of the MS